# Outcomes After VATS Single Versus Multiple Segmentectomy for cT1N0 Non-Small-Cell Lung Cancer

**DOI:** 10.3390/cancers17233814

**Published:** 2025-11-28

**Authors:** Ye Tian, Edoardo Zanfrini, Etienne Abdelnour-Berchtold, Matthieu Zellweger, Jean Yannis Perentes, Thorsten Krueger, Michel Gonzalez

**Affiliations:** 1Faculty of Biology and Medicine, University of Lausanne (UNIL), 1011 Lausanne, Switzerland; ye.tian@unil.ch (Y.T.); jean.perentes@chuv.ch (J.Y.P.); thorsten.krueger@chuv.ch (T.K.); 2Department of Thoracic Surgery, Lausanne University Hospital (CHUV), 1011 Lausanne, Switzerland; edoardo.zanfrini@chuv.ch (E.Z.); etienne.abdelnour@chuv.ch (E.A.-B.); matthieu.zellweger@chuv.ch (M.Z.)

**Keywords:** segmentectomy, sublobar resection, video-assisted thoracoscopic surgery (VATS), non-small cell lung cancer, minimal invasive surgery

## Abstract

This study aimed to compare the short- and long-term outcomes of video-assisted thoracoscopic surgery (VATS) single segmentectomy (SS) versus multiple segmentectomy (MS) for early-stage non-small cell lung cancer (NSCLC) measuring ≤ 3 cm. We conducted a retrospective study of patients with cT1N0 NSCLC. SS was associated with shorter operative time, reduced drainage duration, shorter hospital length of stay, and a lower rate of atrial fibrillation compared with MS. Lymph node dissection and oncological outcomes were comparable. We do not observe significant differences between SS and MS in overall survival, disease-free survival, or local recurrence. The findings indicate that SS and MS provide equivalent cancer control, while SS offers better perioperative recovery. SS may therefore be preferred when adequate margins can be achieved, supporting a more lung-sparing surgical approach in early-stage NSCLC.

## 1. Introduction

Lung cancer remains one of the leading causes of cancer-related morbidity and mortality worldwide, with non-small cell lung cancer (NSCLC) accounting for the majority of cases [1]. Surgical resection remains the standard curative approach for early-stage disease in medically operable patients. For decades, lobectomy with systematic lymph node dissection has been regarded as the standard [2]. However, recent three randomized trials (JCOG0802/WJOG4607L, CALGB 140503, and the German DRKS4897 trial) have demonstrated that anatomical segmentectomy provides oncologic outcomes comparable to, or even superior to, lobectomy for tumors ≤ 2 cm, while offering improved preservation of pulmonary function [3,4,5]. These findings have led to a substantial increase in the use of anatomical segmentectomy in clinical practice [6].

While segmentectomy is now widely accepted for cT1a tumors (≤2 cm), its role in tumors measuring between 2 and 3 cm (cT1b–c) remains less clearly defined [7]. Several retrospective studies and registry analyses have suggested that segmentectomy may be oncologically acceptable in selected 2–3 cm tumors, particularly those with ground-glass dominant or peripherally located [8,9,10,11,12,13,14,15,16]. However, concerns persist regarding the risk of local recurrence when segmentectomy is applied to larger tumors, especially in cases located near segmental boundaries where achieving sufficient margin distance may be challenging. [7]

In this context, the extent of segmental resection becomes a key consideration. Anatomical segmentectomy is not a uniform procedure [17]. It is important to differentiate the extent of resection (single vs. multiple segments) from the technical complexity of the procedure [17,18]. Complex segmentectomies are defined by the need to divide non-linear intersegmental planes or and such complexity may occur in either single or multiple segmentectomies [19,20,21]. Although multiple segmentectomies (MS) involve the removal of two or more contiguous segments, they may follow relatively linear intersegmental boundaries when the tumor straddles adjacent segments. In contrast, many single segmentectomies (SS) require dissection along several intersegmental planes. Consequently, SS can be technically more demanding than MS in selected anatomical regions, despite involving a smaller volume of lung parenchyma [22]. This complexity may influence operative time, risk of prolonged air leak, and the ability to achieve an adequate resection margin [23]. However, whether SS and MS provide equivalent oncologic outcomes remains uncertain. Several studies have reported that MS, by removing a larger parenchymal volume and potentially ensuring wider resection margins, may reduce the risk of local recurrence [24,25,26]. A recent retrospective study reported no significant difference in recurrence or survival between single-segment and multi-segment resections for stage IA tumors ≤ 2 cm [27]. However, that analysis included few complex single segmentectomies, limiting its applicability to cases in which SS requires anatomically challenging plane identification. The comparative value of SS versus MS—particularly for tumors approaching 3 cm—therefore remains unresolved.

The objective of this study was to compare perioperative and oncologic outcomes of video-assisted thoracoscopic surgery (VATS) single segmentectomy versus multiple segmentectomy in patients with cT1N0 NSCLC tumors ≤ 3 cm.

## 2. Materials and Methods

### 2.1. Ethical Statement

The study was conducted in accordance with the Declaration of Helsinki (as revised in 2013). This study was approved by the local ethical committee (CER-VD in Lausanne, Switzerland; referral number: N°2025-01123 approved on the 5 June 2025).

### 2.2. Study Design and Patient Selection

We conducted a retrospective, single-center observational study of consecutive patients with cT1 NSCLC who underwent video-assisted thoracoscopic surgery (VATS) single (SS) or multiple segmentectomy (MS) with mediastinal lymphadenectomy at Lausanne University Hospital between January 2017 and December 2022. All procedures were performed by one of four board-certified thoracic surgeons with extensive experience in VATS anatomical resections (>100 cases).

Eligible patients were adults (≥18 years) with written informed general consent that underwent planned R0 resection pulmonary segmentectomy with systematic lymph node dissection. Only tumors staged as cT1 according to the TNM classification system (8th edition) with a maximum diameter of 3 cm were included (Figure 1). We qualified only specific histological subtypes of NSCLC including adenocarcinoma, squamous cell carcinoma, large-cell carcinoma, and adenosquamous carcinoma. Preoperative evaluation within 30 days of surgery consisted of contrast-enhanced chest CT and FDG-PET/CT. Patients were excluded if they underwent alternative anatomic or non-anatomic resections, such as wedge resection, bilobectomy, sleeve lobectomy, pneumonectomy or open thoracotomy. Patients with synchronous tumors, multifocal disease, or non-qualifying histological subtypes such as carcinoid or small-cell carcinoma were excluded. Patients with a history of ipsilateral thoracotomy, chemotherapy, or radiotherapy were not eligible.

### 2.3. Data Collection

Demographic, clinical, operative, and oncological data were extracted from the institutional electronic database. Variables included age, sex, smoking history, pulmonary function tests, ASA score, tumor size and location (central vs. peripheral), histological type, surgical margin distance, number of lymph nodes dissected, operative time, surgical approach, postoperative complications, drainage duration, length of stay, recurrence, and survival status. A peripheral lesion was defined as situated in the outer one-third of the lung, adjacent to the visceral pleura, whereas central lesion was located within the inner two-thirds, proximate to the bronchi and great vessels of the hilar region.

The primary outcomes were overall survival (OS), disease-free survival (DFS), and local recurrence-free survival (LRFS). OS was defined as the interval from surgery to death from any cause or last follow-up. DFS was defined as the time from surgery to recurrence, metastasis, death, or last follow-up. LRFS was calculated from surgery to the date of local or regional recurrence. A competing risk model was used to evaluate lung cancer–specific mortality and recurrence while accounting for non-cancer deaths. Secondary outcomes included perioperative mortality within 30 days and postoperative morbidity, with particular attention to major complications such as atrial fibrillation, pneumonia, or prolonged air leak. Postoperative complications were classified by organ system. No severity grading system was applied.

### 2.4. Surgical Approach and Workup

All cases were discussed preoperatively at a multidisciplinary tumor board. Standard staging included CT, FDG-PET/CT with SUV assessment, and tissue diagnosis by transthoracic or bronchoscopic biopsy when feasible. In cases with suspected nodal involvement, endobronchial ultrasound with fine-needle aspiration or mediastinoscopy was performed prior to surgery. Anatomical segmentectomy was defined as the systematic dissection and division of the segmental bronchus, artery, and vein, followed by identification and division of the intersegmental plane. Single segmentectomy (SS) was defined as the resection of one bronchopulmonary segment, whereas multiple segmentectomy (MS) was defined as the resection of two or more contiguous anatomical segments. We additionally classified segmentectomies as simple or complex according to the number and configuration of intersegmental planes that must be divided. Simple segmentectomies require the division of a single intersegmental plane and include left upper division segmentectomy or lingulectomy, apical segmentectomy (S6), and basal segmentectomy (S7–10). In contrast, complex segmentectomies involve the division of multiple or irregular intersegmental planes, such as individual or combined segmentectomies in the upper lobes, middle lobe, or selected basal segments.

The choice between SS and MS was based on tumor characteristics, anatomical considerations, and the feasibility of achieving oncologically adequate margins. The decision to perform single (SS) or multiple segmentectomy (MS) was based on preoperative thin-slice CT assessment and intraoperative evaluation of segmental anatomy. SS was performed when the tumor was clearly confined within a single segment and a parenchymal margin ≥ 2 cm or ≥tumor diameter could be safely achieved along one intersegmental plane. MS was chosen when the tumor was located near or across a segmental border, when achieving an adequate margin within one segment was not feasible.

A uniportal or triportal VATS approach was employed according to surgeon preference. Segmentectomy was performed with dissection of the segmental bronchus, artery, and vein, followed by systematic hilar and mediastinal lymph node dissection. Anatomical planning was based on high-resolution CT imaging. For small or deep lesions requiring localization, CT-guided hookwire placement was selectively performed prior to surgery. Intersegmental planes were identified with intravenous indocyanine green fluorescence when necessary and divided using staplers. Frozen section was not routinely performed to assess parenchymal margins, except in cases of suspected hilar nodal involvement, where N1 upstaging prompted conversion to a larger resection.

All cases were postoperatively discussed by a multidisciplinary tumor board to determine the necessity of adjuvant treatment. Follow-up included chest CT scans conducted every three months for the first two years, followed by scans every six months for a total duration of five years. Local recurrence was defined as tumour reappearance at the bronchial or parenchymal resection site, or within ipsilateral N1 lymph nodes, including peribronchial, hilar, or intrapulmonary stations. Distant recurrence was defined as metastatic dissemination to N2 lymph nodes (mediastinal or subcarinal), other lung lobes, the pleura, or extra-thoracic organs. Patient inclusion ended in December 2022 to ensure at least 24 months of potential follow-up to 31 December 2024.

### 2.5. Statistical Analysis

Missing data rates for all variables ranged from 0% to 6.1%. The highest rate was observed for the Charlson Comorbidity Index (6.1%), followed by PET SUV (5.0%), DLCO and margin distance (3.9% each), C/T ratio, tumor location, and pleural invasion (3.4% each). Drain duration had a missing rate of 1.1%, while BMI and FEV1 were missing in 0.8% of cases. Tumor size had the lowest missing rate (0.3%). To minimize bias and preserve statistical power, we utilized statistical imputation methods. The Kolmogorov–Smirnov and Shapiro–Wilk tests confirmed the abnormal distribution of all continuous variables. Continuous variables were compared using independent *t*-tests for normally distributed data or Mann–Whitney U tests for non-normally distributed data. Categorical variables were assessed with Chi-square or Fisher’s exact tests as appropriate. Results are reported as mean ± standard deviation, median with interquartile range, or frequency with percentage. The primary survival endpoints were overall survival (OS), disease-free survival (DFS), and local recurrence-free survival (LRFS). Survival distributions were estimated using the Kaplan–Meier method, and intergroup comparisons were performed using log-rank tests. Survival probabilities with 95% confidence intervals (95% CI) are reported at relevant time points. To account for competing risks, Fine–Gray subdistribution hazard models were applied for analysis of locoregional recurrence, with distant metastasis and non-cancer-related deaths considered as competing events. Cumulative incidence functions (CIFs) were estimated and compared between SS and MS groups using Gray’s test. Sensitivity analyses were performed to confirm the robustness of the findings. To reduce selection bias, we performed a sensitivity analysis using propensity score matching (PSM). Propensity scores were estimated using a multivariable logistic regression model including nine baseline covariates: age, smoking status, Charlson Comorbidity Index, comorbidities, FEV1, tumor size, consolidation-to-tumor ratio, involved lobe, and tumor location. Patients undergoing SS and MS were matched 1:1 using nearest-neighbor matching with a caliper of 0.3 SD of the logit of the propensity score. Covariate balance was assessed using standardized mean differences (SMD), with <0.1 indicating adequate balance. Survival outcomes (OS, DFS, LRFS) in the matched cohort were evaluated using Kaplan–Meier curves and log-rank tests. PSM results are provided in the Appendix A. A two-sided *p*-value of less than 0.05 was considered statistically significant. All statistical analyses were conducted using R Statistical Software (Version 4.2.2, http://www.R-project.org, accessed on 10 August 2025, The R Foundation) and the Free Statistics analysis platform (Version 2.3, Beijing, China).

## 3. Results

Between 2017 and 2022, 334 patients with cT1N0 NSCLC underwent VATS segmentectomy at Lausanne University Hospital. Among them, 211 patients (63%) underwent single segmentectomy (SS) and 123 (37%) multiple segmentectomy (MS, ≥2 segments). The annual distribution of segmentectomy and lobectomy procedures performed at our institution between 2017 and 2022 for cT1 N0 NSCLC is shown in Appendix A. Over this period, segmentectomy progressively increased and ultimately surpassed lobectomy in early-stage NSCLC.

Baseline comorbidity (CCI, ASA) and pulmonary function (FEV1 and DLCO) were similar between groups. (Table 1). The mean tumor diameter for the entire cohort was 14.7 ± 6.3 mm, with no significant difference between SS and MS (14.3 vs. 15.5 mm; *p* = 0.115). The majority of lesions were peripherally located (70%), and adenocarcinoma accounted for over 80% of cases.

Operative characteristics are summarized in Table 2. SS was associated with shorter operative time (117 ± 46 vs. 132 ± 52 min; *p* = 0.007). Median parenchymal margin distance and margin-to-tumor ratio were significantly larger in SS compared to MS (13 vs. 11 mm; *p* = 0.038) (0.9 vs.0.7; *p* = 0.029). The number of hilar and mediastinal lymph nodes harvested did not differ (between groups 8.0 vs. 8.8; *p* = 0.194). R0 resection was achieved in 99.7% of patients, with only one case of microscopically positive margin. In the SS group, 83 (39%) were simple and 128 (61%) complex resections, whereas in the MS group, 67 (54%) were simple and 56 (46%) complex (Figure 2). Upper lobe lesions were more frequent in SS, whereas lower lobe tumors were proportionally higher in MS (*p* < 0.001). Complex procedures were more frequent in the SS with predominant upper lobe segmentectomies, while simple procedures predominated in MS, particularly left upper division resections.

The overall complication profile was similar between groups, except for a higher rate of atrial fibrillation in MS (5.7% vs. 1.4%; *p* = 0.042) (Table 3). Pneumonia and prolonged air leak were numerically more frequent in MS but did not reach statistical significance. The length of postoperative drainage was significantly shorter in the SS group compared to the MS group (1 day vs. 3 days; *p* < 0.001). Median length of stay was longer in MS (6 vs. 5 days; *p* < 0.001). Perioperative mortality was 0%.

With a median follow-up of 30.1 months (IQR: 9–46), we did not observe significant survival or disease-free survival between groups. Five-year overall survival was 94.5% (95% CI; 89.8–99.5%) for SS and 90.7% (78.1–100%) for MS (*p* = 0.23) (Figure 3a). Five-year disease-free survival was 83.2% (77.1–89.8%) vs. 79.1% (70.3–88.9%) (*p* = 0.91) (Figure 3b). Local recurrence-free survival was also equivalent, with 5-year rates of 95.5% (93.1–98.2%) vs. 92.5% (88.1–97.1%) (*p* = 0.44) (Figure 3c).

At 36 months, competing risk analysis did not show a significant difference with the cumulative incidence of local recurrence of 5.5% for SS vs. 6.1% for MS (*p* = 0.394) (Figure 3d) (Table 4). Competing events (distant recurrence or non-cancer death) were slightly higher in MS (13.9% vs. 9.1%, *p* = 0.368), but not statistically significant.

The propensity score matching analysis produced a matched cohort of 148 patients (74 SS and 74 MS). Baseline characteristics were well balanced across all nine covariates (SMD < 0.1). In the matched cohort, OS (*p* = 0.414), DFS (*p* = 0.700), and LRFS (*p* = 0.826) did not differ significantly between SS and MS. A full summary of PSM-adjusted survival outcomes is presented in Appendix A.

## 4. Discussion

Our study evaluated outcomes of VATS SS versus MS in 334 patients with cT1N0 NSCLC. Both approaches were associated with excellent perioperative outcomes, with no mortality. Compared with MS, SS was associated with shorter operative times, fewer arrhythmias, and reduced length of stay, while oncological outcomes, including OS, DFS and LRFS were equivalent. Competing risk analysis confirmed no significant differences in local recurrence or other events. Our results suggest that SS provides comparable oncological efficacy to MS while offering better perioperative outcomes.

Segmentectomy has emerged as a valid option for early-stage peripheral NSCLC of less than 2 cm. Recent randomized trials have demonstrated non-inferior oncological outcomes compared with lobectomy, while offering superior pulmonary function preservation [3,4,5]. These findings have reshaped surgical practice, expanding the role of sublobar resections in selected early-stage patients and stimulating investigation into their applicability to slightly larger tumors (2–3 cm).

Despite growing acceptance of segmentectomy, uncertainty persists regarding the optimal extent of resection (SS or MS) for early-stage lung cancer. The rationale for MS lies in achieving wider parenchymal margins and potentially improving local control, but this may come at the expense of greater functional loss and perioperative morbidity [24,25,26]. Conversely, SS may be technically more challenging but spares more lung parenchyma and concerns remain about sufficient surgical margin. Few studies have directly compared SS and MS. In the recent study by Al-Thani et al., comparing SS and MS resections in clinical stage IA NSCLC ≤ 2 cm, no significant difference was observed in locoregional recurrence (6.3% vs. 7.4%; *p* = 0.72) or overall survival (87% vs. 84%) despite smaller tumors in the SS group and a non-significant difference in margin length (1.7 cm vs. 2.0 cm; *p* = 0.15) [27]. Our results extend these findings to patients with cT1cN0 tumors up to 3 cm, showing no significant survival or recurrence advantage with MS. Importantly, our series included only minimally invasive VATS segmentectomies, reducing heterogeneity compared with prior studies that included open procedures.

Tumor size, location, and morphology remain central to determining resection type. In our series, upper lobe lesions were more often treated with SS, while lower lobe tumors frequently required MS due to anatomical constraints and margin considerations. Previous studies have emphasized that central tumors or those with higher consolidation-to-tumor ratios (CTR) may warrant MS to achieve adequate margins [12,28]. Conversely, peripheral ground-glass-predominant lesions are well suited for SS [29,30,31]. Margin adequacy, particularly the ratio of margin distance to tumor size, has repeatedly been shown to be a stronger predictor of local control than the number of segments resected [32]. The fact that SS demonstrated a slightly larger median parenchymal margin (13 vs. 11 mm; *p* = 0.038) and margin-to-tumor ratio (0.9 vs. 0.7; *p* = 0.027) compared with MS may appear counterintuitive, but this finding may probably reflect anatomical selection rather than more extensive resection. SS was primarily performed for tumors located well within the boundaries of a single anatomical segment, where the intersegmental plane is regular, and the resection margin can be shaped generously. In contrast, MS was more often required when tumors were positioned near intersegmental borders, where multiple planes intersect, and the resection margin is constrained by neighboring bronchovascular structures. Therefore, even though MS removes a greater volume of lung tissue, the achievable margin may be smaller due to the irregularity of intersegmental boundaries and the need to preserve essential vascular and airway anatomy.

Our study highlights significant perioperative advantages of SS. Operative time was significantly shorter (117.4 ± 45.6 min vs. 132.3 ± 52.4 min; *p* = 0.007), hospital stay (5 vs. 6 days, *p* < 0.001) and length of drainage (1 vs. 3 days; *p* < 0.001) were reduced, and atrial fibrillation (1.4% vs. 5.7%, *p* = 0.042) was less common compared with MS. These findings are consistent with prior reports showing higher complication rates with MS due to longer operative times and more complex dissection [25,26]. While overall morbidity was low in both groups, the simplicity of SS translates into less physiological stress, supporting its preferential use when oncologic criteria are met. In our cohort, a considerable proportion of SS involved segments traditionally considered technically demanding, such as S1, S3, S8, or S9, where intersegmental planes are frequently deep and irregular. Despite this higher anatomical complexity, perioperative outcomes (including air leak, complications, and length of stay) were comparable between SS and MS suggesting that complexity alone does not necessarily translate into increased morbidity. Despite perioperative differences, long-term survival and recurrence patterns were equivalent between SS and MS. Five-year OS and DFS rates exceeded 80% in both groups, and LRFS was >90%. Although Kaplan–Meier and competing risk analyses suggested a nonsignificant trend toward slightly higher local recurrence with SS, this was offset by more competing events (non-cancer death, distant metastasis) in the MS group. The PSM sensitivity analysis further confirmed that the type of segmentectomy was not associated with differences in OS, DFS, or LRFS after balancing nine clinically relevant confounders. These findings align with both Al-Thani et al. [27] and Mathey-Andrews et al. [16], who reported that segmentectomy—including in tumors up to 3 cm—was not oncologically inferior to more extensive resections.

This study has several limitations that should be acknowledged. First, its retrospective single-center design inherently introduces potential selection bias and limits external generalizability. Although baseline characteristics were balanced between groups, unmeasured confounding factors, such as surgeon preference or intraoperative judgment, may have influenced the choice between single and multiple segmentectomy. However, tumor location (central vs. peripheral) did not significantly differ between groups, suggesting that the extent of resection was not systematically driven by anatomical bias. Therefore, the observed differences in perioperative outcomes and margin distance are more likely to reflect differences in procedural complexity rather than underlying tumor distribution. Second, the median follow-up of 30 months is relatively short for assessing long-term oncologic outcomes in NSCLC, particularly regarding late local or distant recurrences. Due to the limited number of patients remaining at risk beyond 48 months, five-year survival estimates would be statistically unreliable and therefore were not reported. Longer surveillance is required to confirm the sustained equivalence observed between groups. Third, the number of oncologic events was limited, resulting in modest statistical power to detect small differences in survival or recurrence. Although competing-risk analyses were performed to strengthen robustness, some comparisons remain exploratory. Fourth, intraoperative frozen section analysis was not routinely performed to verify parenchymal margins. This may have led to underestimation of margin-related local recurrence risk, even though overall R0 resection rates were high. Fifth, postoperative complications were not graded using standardized severity scales such as the Clavien–Dindo classification. Consequently, the true impact of SS and MS on morbidity burden could not be fully quantified. Sixth, postoperative pulmonary function tests and quality-of-life assessments were not systematically collected during the study period and are not currently obtained in a standardized prospective manner at our institution. Consequently, we were unable to evaluate the functional impact of SS versus MS, which represents an inherent limitation. Seventh, we were unable to determine the rate of conversion from planned segmentectomy to lobectomy because the operative records did not uniformly distinguish between an intended segmentectomy that was converted intraoperatively and a planned lobectomy performed from the outset representing an inherent limitation of the retrospective design. Finally, the study reflects the experience of a high-volume VATS center with substantial technical expertise, which may limit reproducibility in lower-volume or less specialized settings.

## 5. Conclusions

In patients with cT1cN0 NSCLC ≤ 3 cm, VATS SS and MS achieved equivalent short-term oncological outcomes. SS was associated with shorter operative time, fewer complications, and shorter hospitalization, supporting its preferential use when adequate margins and lymph node dissection are achievable. While SS and MS demonstrated comparable short-term oncologic outcomes in this study, these findings should be interpreted with caution due to the relatively short follow-up period. Longer-term data are required to confirm durable oncologic equivalence. Our findings reinforce that surgical planning should prioritize oncological clearance and parenchymal preservation, rather than assuming that more extensive segmental resection inherently confers superior oncological benefit. Prospective multicenter studies, ideally integrating three-dimensional reconstruction and intraoperative navigation, are needed to validate these results and further individualize the choice of segmentectomy extent.

## Figures and Tables

**Figure 1 cancers-17-03814-f001:**
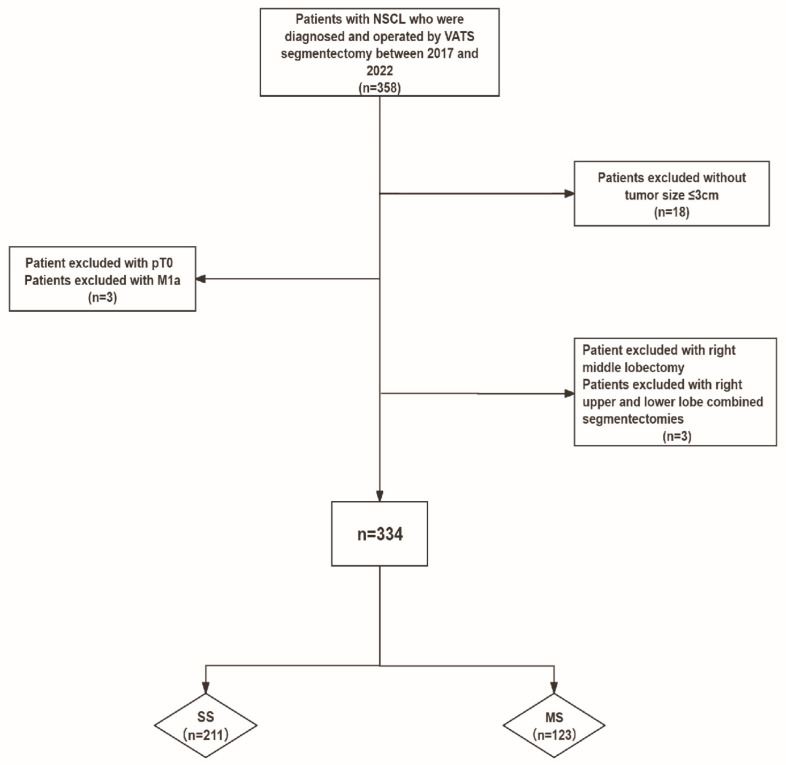
Flowchart. Patients were stratified into two groups based on resection extent: SS or MS.

**Figure 2 cancers-17-03814-f002:**
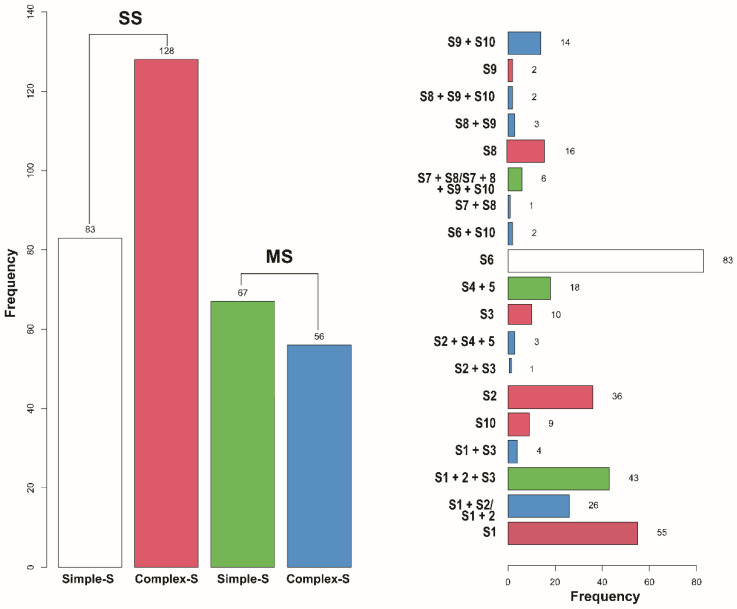
Distribution of Segmentectomy Difficulty by Frequency.

**Figure 3 cancers-17-03814-f003:**
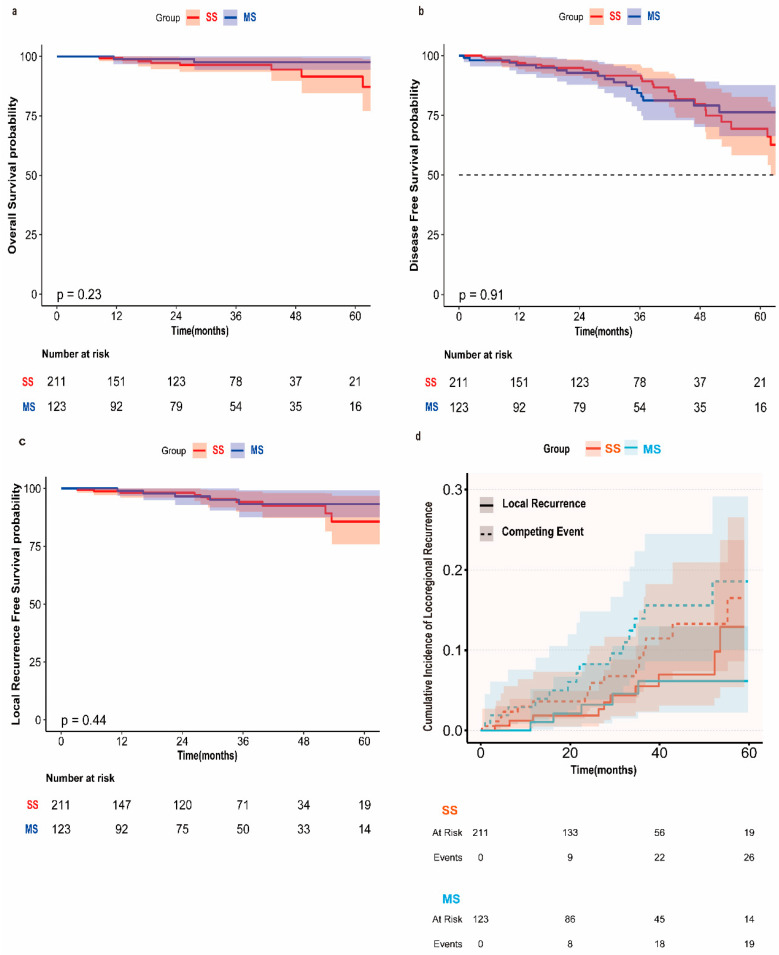
Kaplan–Meier curves of (**a**) overall survival, (**b**) disease-free survival and (**c**) local recurrence free survival, stratified by Single-Segmentectomy (SS) and Multiple Segmentectomy (MS). A competing risk analysis of (**d**) cumulative incidence local recurrence was performed with non-cancer death and distant metastasis defined as competing events.

**Table 1 cancers-17-03814-t001:** Patient Characteristics.

Variables	Total (*n* = 334)	SS (*n* = 211)	MS (*n* = 123)	*p*-Value
Age, Mean ± SD	67.7 ± 9.3	68.0 ± 9.3	67.3 ± 9.3	0.528
Gender, n (%)				0.698
Male	161 (48.2)	100 (47.4)	61 (49.6)	
Female	173 (51.8)	111 (52.6)	62 (50.4)	
Smoking History, n (%)	286 (85.6)	182 (86.3)	104 (84.6)	0.669
BMI, Mean ± SD	25.6 ± 4.9	25.7 ± 4.9	25.5 ± 4.9	0.746
Previous Cancer, n (%)	158 (47.3)	97 (46)	61 (49.6)	0.523
CCI, Mean ± SD	5.3 ± 2.0	5.3 ± 1.9	5.3 ± 2.0	0.922
Comorbidity, n (%)	247 (74.0)	156 (73.9)	91 (74)	0.992
COPD, n (%)	143 (42.8)	87 (41.2)	56 (45.5)	0.444
Hypertension, n (%)	175 (52.4)	110 (52.1)	65 (52.8)	0.9
Cardiovascular Disease, n (%)	101 (30.2)	67 (31.8)	34 (27.6)	0.43
Atrial Fibrillation, n (%)	37 (11.1)	22 (10.4)	15 (12.2)	0.619
Diabetes Mellitus, n (%)	53 (15.9)	30 (14.2)	23 (18.7)	0.28
Renal Insufficiency, n (%)	39 (11.7)	26 (12.3)	13 (10.6)	0.63
FEV1% Predicted, n (%)				0.658
FEV1% Predicted < 80%	136 (40.7)	84 (39.8)	52 (42.3)	
Median FEV1 (%) (IQR)	85.0 (69.0, 100.0)	86.0 (70.0, 100.0)	84.0 (65.0, 99.0)	0.194
DLCO% Predicted, n (%)				0.827
DLCO% Predicted < 80%	212 (63.5)	133 (63)	79 (64.2)	
Median DLCO (%) (IQR)	70.0 (57.0, 84.8)	70.0 (57.5, 84.0)	69.0 (54.5, 86.5)	0.51
ASA, n (%)				0.47 *
ASAI	2 (0.6)	1 (0.5)	1 (0.8)	
ASAII	151 (45.2)	100 (47.4)	51 (41.5)	
ASAIII	176 (52.7)	108 (51.2)	68 (55.3)	
ASAIV	5 (1.5)	2 (0.9)	3 (2.4)	

* Fisher’s exact test;. SS = Single segmentectomy; MS = Multiple segmentectomy; BMI = Body mass index; CCI = Charlson Comorbidity Index; COPD = Chronic obstructive pulmonary disease; FEV1 = Forced expiratory volume in 1 s; DLCO = Diffusing capacity for carbon monoxide; ASA = American Society of Anesthesiologists.

**Table 2 cancers-17-03814-t002:** Surgical Characteristics and Pathological Results.

Variables	Total (*n* = 334)	SS (*n* = 211)	MS (*n* = 123)	*p*-Value
Lobe, n (%)				<0.001
RUL	91 (27.2)	74 (35.1)	17 (13.8)	
RLL	82 (24.6)	63 (29.9)	19 (15.4)	
LUL	112 (33.5)	31 (14.7)	81 (65.9)	
LLL	49 (14.7)	43 (20.4)	6 (4.9)	
PET SUV, n (%)				0.731
SUVmax < 2.5	148 (44.3)	95 (45)	53 (43.1)	
C/T Ratio, n (%)				0.506
C/T Ratio < 0.5	65 (19.5)	45 (21.3)	20 (16.3)	
0.5 ≤ C/T Ratio < 1	97 (29.0)	61 (28.9)	36 (29.3)	
C/T Ratio = 1	172 (51.5)	105 (49.8)	67 (54.5)	
Tumor Size(mm), Mean ± SD	14.7 ± 6.3	14.3 ± 6.0	15.5 ± 6.7	0.115
Tumor Location, n (%)				0.702
Central	99 (29.6)	61 (28.9)	38 (30.9)	
Peripheral	235 (70.4)	150 (71.1)	85 (69.1)	
Margin Distance(mm), Median (IQR)	12.0 (6.0, 20.0)	13.0 (6.5, 22.0)	11.0 (5.0, 20.0)	0.038
Margin-to-tumor ratio; Median (IQR)	0.9 (0.4–1.7)	0.9 (0.5–1.8)	0.7 (0.3–1.5)	0.029
Lymph Nodes Harvested, Mean ± SD	8.3 ± 5.3	8.0 ± 5.5	8.8 ± 5.1	0.194
Operative Time, Mean ± SD	122.9 ± 48.7	117.4 ± 45.6	132.3 ± 52.4	0.007
Conversion open thoracotomy, n (%)	6 (1.8)	1 (0.5)	5 (4)	0.027 *
Resection Status, n (%)				1 *
R0	333 (99.7)	210 (99.5)	123 (100)	
R1	1 (0.3)	1 (0.5)	0 (0)	
Histology, n (%)				0.603
Adenocarcinoma	274 (82.0)	171 (81)	103 (83.7)	
Squamous cell carcinoma	43 (12.9)	30 (14.2)	13 (10.6)	
Others	17 (5.1)	10 (4.7)	7 (5.7)	
Pleural invasion, n (%)	40 (12.0)	24 (11.4)	16 (13)	0.657
T Stage, n (%)				0.1 *
Tis	28 (8.4)	23 (10.9)	5 (4.1)	
T1a	83 (24.9)	50 (23.7)	33 (26.8)	
T1b	131 (39.2)	85 (40.3)	46 (37.4)	
T1c	45 (13.5)	24 (11.4)	21 (17.1)	
T2a	36 (10.8)	20 (9.5)	16 (13)	
T3	11 (3.3)	9 (4.3)	2 (1.6)	
N Stage, n (%)				0.287 *
N0	315 (94.3)	202 (95.7)	113 (91.9)	
N1	7 (2.1)	2 (0.9)	5 (4.1)	
N2	7 (2.1)	4 (1.9)	3 (2.4)	
Nx	5 (1.5)	3 (1.4)	2 (1.6)	
Pathologic Stage, n (%)				0.29 *
Stage 0	28 (8.4)	23 (10.9)	5 (4.1)	
Stage IA1	80 (24.0)	49 (23.2)	31 (25.2)	
Stage IA2	125 (37.4)	82 (38.9)	43 (35)	
Stage IA3	43 (12.9)	24 (11.4)	19 (15.4)	
Stage IB	33 (9.9)	18 (8.5)	15 (12.2)	
Stage IIB	18 (5.4)	11 (5.2)	7 (5.7)	
Stage IIIA	7 (2.1)	4 (1.9)	3 (2.4)	
Adjuvant Chemotherapy, n (%)	49 (14.7)	32 (15.2)	17 (13.8)	0.738

* Fisher’s exact test; mm: millimeter. SS = Single segmentectomy; MS = Multiple segmentectomy; PET SUV = Positron emission tomography standardized uptake value; C/T ratio = Consolidation-to-tumor ratio; IQR = Interquartile range; mm = millimeter.

**Table 3 cancers-17-03814-t003:** Morbidity and Mortality.

Variables	Total (*n* = 334)	SS (*n* = 211)	MS(*n* = 123)	*p*-Value
Pneumonia, n (%)	28 (8.4)	13 (6.2)	15 (12.2)	0.055
Pulmonary Air Leak, n (%)	31 (9.3)	15 (7.1)	16 (13)	0.073
Empyema, n (%)	2 (0.6)	2 (0.9)	0 (0)	0.533 *
Embolism, n (%)	2 (0.6)	0 (0)	2 (1.6)	0.135 *
Atelectasis, n (%)	4 (1.2)	1 (0.5)	3 (2.4)	0.143 *
Arrhythmia, n (%)	10 (3.0)	3 (1.4)	7 (5.7)	0.042 *
Myocardial Infarction, n (%)	0 (0.0)	0 (0.0)	0 (0.0)	1 *
Ileus, n (%)	4 (1.2)	2 (0.9)	2 (1.6)	0.627 *
Colitis, n (%)	1 (0.3)	0 (0)	1 (0.8)	0.368 *
Urosepsis, n (%)	0 (0.0)	0 (0.0)	0 (0.0)	1 *
AKI, n (%)	12 (3.6)	6 (2.8)	6 (4.9)	0.37 *
TIA/Stroke, n (%)	1 (0.3)	1 (0.5)	0 (0)	1 *
Reoperation, n (%)	12 (3.6)	7 (3.3)	5 (4.1)	0.765 *
Drain duration, n (%)				0.701
>5 days	51 (15.3)	31 (14.7)	20 (16.3)	
Median length of drainage days (IQR)	2.0 (1.0, 4.0)	1.0 (1.0, 3.0)	3.0 (1.0, 5.0)	<0.001
Length of Stay Days, Median (IQR)	5.0 (4.0, 8.0)	5.0 (3.0, 7.0)	6.0 (5.0, 9.5)	<0.001
Recurrence, n (%)				0.969
Local Recurrence	9 (2.7)	6 (2.8)	3 (2.4)	
Distant Recurrence	13 (3.9)	9 (4.3)	4 (3.3)	
Local combined Distant Recurrence	7 (2.1)	5 (2.4)	2 (1.6)	
Perioperative Mortality ≤ 30 days	0 (0.0)	0 (0.0)	0 (0.0)	NA

* Fisher’s exact test; SS = Single segmentectomy; MS = Multiple segmentectomy; AKI = Acute kidney injury; TIA = Transient ischemic attack; IQR = Interquartile range.

**Table 4 cancers-17-03814-t004:** Comparison of Cumulative Incidence Between SS and MS Groups and Gray Test Results.

Event Type	Group	12-Month Cumulative Incidence (%) (95% CI)	36-Month Cumulative Incidence (%) (95% CI)	At-Risk Population (12/36 Months)	Gray Test χ^2^	*p*-Value
Local Recurrence	SS	1.85 (0.50–4.93)	5.51 (2.36–10.62)	148/71	0.73	0.394
Local Recurrence	MS	1.04 (0.09–5.14)	6.14 (2.21–12.96)	93/50	-	-
Competing Event	SS	3.60 (1.48–7.28)	9.06 (4.79–14.98)	149/72	0.81	0.368
Competing Event	MS	2.89 (0.77–7.56)	13.94 (7.49–22.35)	93/51	-	-

Competing Event: non-cancer death or distant recurrence. SS = Single segmentectomy; MS = Multiple segmentectomy; CI = Confidence interval.

## Data Availability

The data are available from the corresponding author upon reasonable request.

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
