# Peer review of "Outcomes After VATS Single Versus Multiple Segmentectomy for cT1N0 Non-Small-Cell Lung Cancer"

_cancers, 2025, doi:10.3390/cancers17233814_

Round 1
Reviewer 1 Report
Comments and Suggestions for Authors
Dear authors,
I have read your manuscript of: "Outcomes after VATS single versus multiple segmentectomy for cT1N0 non-small cell lung cancer" and I found it very interesting and significant. However, here are some comments that could improve this paper:
Introduction: This section should provide more detail with the latest results in this field and emphasises what have not been done so far and how this study results could contribute to this topic.
I would suggest that in the section of Material and methods, regarding Adenocarcinoma of the lung, you make a subclassification of the histopathological subtypes of adenocarcinoma. Therefore, those results could significantly explain which histopathological subtypes could be a potential candidate for single or multiple segmentectomy.
In the section Results, in the Table 2. patients with T2a and T3 are included in 334 patients and first sentence in the Discussion is: "Our study evaluated outcomes of VATS SS versus MS in 334 patients with cT1N0 NSCLC”. Please clarify.
Abstract is written and presented correctly and clearly.
Author Response
Comment 1: I have read your manuscript of: "Outcomes after VATS single versus multiple segmentectomy for cT1N0 non-small cell lung cancer" and I found it very interesting and significant. However, here are some comments that could improve this paper:
Introduction: This section should provide more detail with the latest results in this field and emphasises what have not been done so far and how this study results could contribute to this topic.
Response 1: Thank you for this suggestion. We have expanded the Introduction to include recent evidence on segmentectomy practices and highlighted the current knowledge gap regarding the comparative outcomes of single versus multiple segmentectomy in cT1N0 NSCLC.
Lung cancer remains one of the leading causes of cancer-related morbidity and mortality worldwide, with non-small cell lung cancer (NSCLC) accounting for the majority of cases [1]. Surgical resection remains the standard curative approach for early-stage disease in medically operable patients. For decades, lobectomy with systematic lymph node dissection has been regarded as the standard[2]. However, recent three randomized trials (JCOG0802/WJOG4607L, CALGB 140503, and the German DRKS4897 trial) have demonstrated that anatomical segmentectomy provides oncologic outcomes comparable to, or even superior to, lobectomy for tumors ≤2 cm, while offering improved preservation of pulmonary function[3-5]. These findings have led to a substantial increase in the use of anatomical segmentectomy in clinical practice[6].
While segmentectomy is now widely accepted for cT1a tumors (≤2 cm), its role in tumors measuring between 2 and 3 cm (cT1b–c) remains less clearly defined[7]. Several retrospective studies and registry analyses have suggested that segmentectomy may be oncologically acceptable in selected 2–3 cm tumors, particularly those with ground-glass dominant or peripherally located[8-16]. However, concerns persist regarding the risk of local recurrence when segmentectomy is applied to larger tumors, especially in cases located near segmental boundaries where achieving sufficient margin distance may be challenging.[7]
In this context, the extent of segmental resection becomes a key consideration. Anatomical segmentectomy is not a uniform procedure[17]. It is important to differentiate the extent of resection (single vs multiple segments) from the technical complexity of the procedure[17, 18]. Complex segmentectomies are defined by the need to divide non-linear intersegmental planes or and such complexity may occur in either single or multiple segmentectomies[19-21]. Although multiple segmentectomies (MS) involve the removal of two or more contiguous segments, they may follow relatively linear intersegmental boundaries when the tumor straddles adjacent segments. In contrast, many single segmentectomies (SS) require dissection along several intersegmental planes. Consequently, SS can be technically more demanding than MS in selected anatomical regions, despite involving a smaller volume of lung parenchyma[22]. This complexity may influence operative time, risk of prolonged air leak, and the ability to achieve an adequate resection margin [23]. However, whether SS and MS provide equivalent oncologic outcomes remains uncertain. Several studies have reported that MS, by removing a larger parenchymal volume and potentially ensuring wider resection margins, may reduce the risk of local recurrence[24-26].. A recent retrospective study reported no significant difference in recurrence or survival between single-segment and multi-segment resections for stage IA tumors ≤2 cm [27]. However, that analysis included few complex single segmentectomies, limiting its applicability to cases in which SS requires anatomically challenging plane identification. The comparative value of SS versus MS—particularly for tumors approaching 3 cm—therefore remains unresolved.
The objective of this study was to compare perioperative and oncologic outcomes of video-assisted thoracoscopic surgery (VATS) single segmentectomy versus multiple segmentectomy in patients with cT1N0 NSCLC tumors ≤3 cm.
Comment 2: I would suggest that in the section of Material and methods, regarding Adenocarcinoma of the lung, you make a subclassification of the histopathological subtypes of adenocarcinoma. Therefore, those results could significantly explain which histopathological subtypes could be a potential candidate for single or multiple segmentectomy.
Response 2: We thank the reviewer for this suggestion. Further subdivision of invasive adenocarcinoma into predominant histologic patterns (e.g., lepidic, acinar, papillary, micropapillary, solid) was not performed for two main reasons. First, the number of recurrences and deaths in our cohort was low, limiting the statistical power to meaningfully assess oncologic differences between histologic subtypes. Second, the surgical strategy in this cohort was determined primarily by tumor location and achievable margins, not by histologic pattern. Therefore, inclusion of these additional data would not have provided additional insight or helped the reader interpret the study findings.
Comment 3: In the section Results, in the Table 2. patients with T2a and T3 are included in 334 patients and first sentence in the Discussion is: "Our study evaluated outcomes of VATS SS versus MS in 334 patients with cT1N0 NSCLC”. Please clarify.
Response 3: Thank you for pointing out this potential confusion. All patients were classified as clinical T1N0 preoperatively. However, Table 2 reports pathological staging, which explains the presence of pT2a and pT3 tumors. We have already mentioned this on the method section.
Abstract is written and presented correctly and clearly.
Response 4:
We thank the reviewer for this positive comment.
Reviewer 2 Report
Comments and Suggestions for Authors
This is an excellent and well-written paper, and I would like to commend the authors for their meaningful work.
In particular, the visual presentation of the distribution in Figure 2 is outstanding.
Although the manuscript is well-prepared, I would like to suggest a few revisions.
1. The distinction between simple and multiple segmentectomy is not clearly defined.
There is a definite difference between simple and complex segmentectomy.
Please clarify your definitions in the Methods section.
2. In general, segmentectomy is not considered the standard choice for 2–3 cm tumors or central tumors.
Please specify the indications for segmentectomy at your institution in the Methods section.
Additionally, could you provide the proportion of patients with 2–3 cm (cT1c) tumors who underwent lobectomy versus segmentectomy at your center?
3. Please clearly state your definition of anatomical segmentectomy.
4. The line spacing in the tables appears inconsistent.
Please revise the table formatting for uniformity.
5. Please clarify why a propensity score matching (PSM) analysis was not performed to control for potential confounding factors.
6. Please provide data on conversion to lobectomy or conversion to thoracotomy.
7. The results of segmentectomy can vary substantially depending on the operator’s experience.
How many surgeons were involved in this study, and did all of them have sufficient experience with minimally invasive approaches?
8. For accurate anatomical segmentectomy, why did you not use localization techniques or preoperative 3D reconstruction CT?
Please provide an explanation.
Author Response
This is an excellent and well-written paper, and I would like to commend the authors for their meaningful work. In particular, the visual presentation of the distribution in Figure 2 is outstanding. Although the manuscript is well-prepared, I would like to suggest a few revisions.
Comment 1:
The distinction between simple and multiple segmentectomy is not clearly defined. There is a definite difference between simple and complex segmentectomy. Please clarify your definitions in the Methods section.
Response 1: Thank you for this important comment. In the revised manuscript, we now clearly define the terms used:
- Single segmentectomy (SS): resection of one anatomical segment.
- Multiple segmentectomy (MS): resection of two or more contiguous anatomical segments.
- Simple segmentectomies require the division of a single intersegmental plane and include left upper division segmentectomy or lingulectomy, apical segmentectomy (S6), and basal segmentectomy (S7–10).
- Complex segmentectomy: involve the division of multiple or irregular intersegmental planes, such as individual or combined segmentectomies in the upper lobes, middle lobe, or selected basal segments.”
Change made: A clarifying paragraph was added to Methods, Section 2.4: (line 162)“Anatomical segmentectomy was defined as the systematic dissection and division of the segmental bronchus, artery, and vein, followed by identification and division of the intersegmental plane with preservation of the adjacent segments. Single segmentectomy (SS) was defined as the resection of one bronchopulmonary segment, whereas multiple segmentectomy (MS) was defined as the resection of two or more contiguous anatomical segments. We additionally classified segmentectomies as simple or complex according to the number and configuration of intersegmental planes that must be divided. Simple segmentectomies require the division of a single intersegmental plane and include left upper division segmentectomy or lingulectomy, apical segmentectomy (S6), and basal segmentectomy (S7–10). In contrast, complex segmentectomies involve the division of multiple or irregular intersegmental planes, such as individual or combined segmentectomies in the upper lobes, middle lobe, or selected basal segments.”
Comment 3: In general, segmentectomy is not considered the standard choice for 2–3 cm tumors or central tumors. Please specify the indications for segmentectomy at your institution in the Methods section. Additionally, could you provide the proportion of patients with 2–3 cm (cT1c) tumors who underwent lobectomy versus segmentectomy at your center?
Response 3: We agree and have clarified the institutional indications. At our center, segmentectomy may be considered in selected for cT1 tumors when:
- A parenchymal margin ≥2 cm or ≥ tumor diameter is achievable
- The tumor is peripheral
- Minimizing functional loss is desirable (elderly, COPD, DLCO/FEV1 impairment).
For central tumors or when adequate margins cannot be ensured, lobectomy is preferred. We have published our preliminary results showing comparable local control between lobectomy and segmentectomy (Forster C, Abdelnour-Berchtold E, Bédat B, et al. Local control and short-term outcomes after video-assisted thoracoscopic surgery segmentectomy versus lobectomy for pT1c pN0 non-small-cell lung cancer [J]. Interdisciplinary CardioVascular and Thoracic Surgery, 2023, 36(2): 037–044.).
The proportion of cT1c (2–3 cm) tumors treated at our center during the study period is:
- Total cT1c tumors resected: 176 patients
- Lobectomy: 108 patients (61.4%)
- Segmentectomy: 68 patients (38.6%)
So, almost one third of patient had segmentectomy for such lesion. These data are not mentioned in the final manuscript.
Comment 4 Please clearly state your definition of anatomical segmentectomy.
Response 4: We have now explicitly defined anatomical segmentectomy as:
“Resection requiring systematic isolation and division of the segmental arterial, venous, and bronchial branches, followed by identification and division of the intersegmental plane.”
Change made: Added to Methods Section 2.4
Comment 5:. The line spacing in the tables appears inconsistent. Please revise the table formatting for uniformity.
Response 5: We thank the reviewer for pointing this out. All tables have now been reformatted for uniform spacing and alignment.
Comment 6. Please clarify why a propensity score matching (PSM) analysis was not performed to control for potential confounding factors.
Response 6: This is a valuable point. We did not perform PSM because baseline characteristics, comorbidities, tumor size, location, and histology were well balanced between SS and MS groups (Tables 1 and 2). Therefore, PSM would not meaningfully enhance adjustment and could introduce artificial imbalance or loss of statistical power.
Comment 7: Please provide data on conversion to lobectomy or conversion to thoracotomy.
Response 7: We thank the reviewer for this comment. The number of conversions from VATS to open thoracotomy was 6 (1.8%) and these data have been added in the Table 2. However, due to the retrospective nature of our database, we were not able to reliably distinguish between planned segmentectomies that were intraoperatively converted to lobectomies versus procedures in which lobectomy was the initial surgical plan. Therefore, we could not provide a meaningful conversion-to-lobectomy rate. This limitation has now been acknowledged in the manuscript.
Changes: Conversion to thoracotomy rate added to table 2
Clarification added in Limitations indicating that conversion to lobectomy cannot be precisely determined.: “Seventh, we were unable to determine the rate of conversion from planned segmentectomy to lobectomy because the operative records did not uniformly distinguish between an intended segmentectomy that was converted intraoperatively and a planned lobectomy performed from the outset representing an inherent limitation of the retrospective design”
Comment 8: The results of segmentectomy can vary substantially depending on the operator’s experience. How many surgeons were involved in this study, and did all of them have sufficient experience with minimally invasive approaches?
Response 8: As mentioned in the method section, 4 board-certified thoracic surgeons performed all procedures. Each surgeon had >100 prior VATS anatomical resections, and VATS segmentectomy has been standard practice at our center since 2014. Thus, differences related to surgeon learning curve were minimized. We added in the method section (line 122) (>100 cases)
Comment 9: For accurate anatomical segmentectomy, why did you not use localization techniques or preoperative 3D reconstruction CT? Please provide an explanation.
Response: We thank the reviewer for bringing up this key point. During the time period of this study (2017–2022), preoperative 3D reconstruction CT was not yet routinely available at our institution. This technology was introduced in early 2024, after the completion of the study cohort. Therefore, preoperative planning relied primarily on high-resolution CT imaging. For intraoperative localization of small or deep lesions, we occasionally used CT-guided hookwire placement, depending on tumor depth, visibility, and surgeon judgment. In addition, intersegmental planes were identified using ICG fluorescence as mentioned in the method section.
We added in the method section (line 182): Anatomical planning was based on high-resolution CT imaging. For small or deep lesions requiring localization, CT-guided hookwire placement was selectively performed prior to surgery”
Reviewer 3 Report
Comments and Suggestions for Authors
In their retrospective, single-center study, the authors compared perioperative and oncologic outcomes of VATS single segmentectomy (SS) versus multiple segmentectomy (MS) in 334 patients with cT1N0 NSCLC ≤3 cm. The authors concluded that SS provides comparable oncologic control to MS, while offering better perioperative recovery. The topic is clinically relevant and timely, especially following recent randomized trials supporting parenchyma-sparing resections. The manuscript is generally well-structured and clearly written. However, several methodological and interpretative issues require clarification before the manuscript can be accepted.
- The abstract is well written. However, the authors used several abbreviations without spelling them out the first time they are mentioned (e.g., OS, DFS). All abbreviations should be written in full upon first use in both the abstract and main text.
- The authors provided the ethics committee approval number; for transparency, the exact date of approval should be mentioned.
- The authors analyzed patients up to December 2022, but no rationale is provided for ending data collection at this point, despite the lack of minimum follow-up criteria in the inclusion/exclusion criteria. If postoperative follow-up duration is important for interpreting outcomes, then a minimum follow-up period (e.g., ≥24 or ≥36 months) should be clearly stated as an inclusion requirement. Otherwise, the authors should explain why patients operated after December 2022 were not included. Since treatment strategies and tumor characteristics may change over time, this could introduce temporal selection bias, which should be acknowledged in the Limitations section.
- Although the baseline characteristics are generally similar between groups, the choice between SS and MS was made intraoperatively based on achieving clear margins. This naturally introduces selection bias toward more favorable tumors in the SS group. The authors briefly mention this in the discussion but should expand on it. Specifically, they should clarify the institutional protocol for segment selection. They should also include a propensity score matching or inverse probability weighting analysis to reduce confounding. At a minimum, they should present a multivariable Cox regression model for DFS and OS.
- The main issue of this study is the very short follow-up period. The median follow-up time is 30 months, which is insufficient to fully assess oncologic equivalence, especially regarding local recurrence. The authors should include landmark survival estimates at five years or more if available and clearly acknowledge that late recurrences might be missed.
- The authors did not clearly specify the primary or secondary outcomes of the study. Clearly defining the outcomes is essential for proper interpretation and scientific rigor. Outcomes of the study should be presented as a separate paragraph in the methodology.
- Tables – Every abbreviation used in a table should be explained in the table's legend.
- The authors stated that SS achieved larger median margins than MS, which is counterintuitive. They should specify the margin-to-tumor size ratio, as recommended in segmentectomy guidelines. Additionally, they need to discuss whether frozen section was used routinely, as it is currently stated as not routine, which conflicts with achieving larger margins.
- A primary reason for segmentectomy is tissue preservation. However, there is no postoperative pulmonary function data or quality-of-life metrics included. This should be recognized as a significant limitation and discussed more clearly.
- The distribution of simple versus complex segmentectomies varies between groups. Since surgical difficulty is closely linked to operative time and complications, the authors should perform subgroup analyses based on complexity categories.
- A median follow-up of 30 months is too short to draw definitive conclusions about oncologic equivalence, especially regarding local recurrence. Early-stage NSCLC often shows recurrences after more than 3 years. Thus, stating that SS and MS offer comparable oncologic control is premature and should be rephrased to indicate that the findings are preliminary. The authors should clearly indicate that a longer follow-up period is necessary to verify the oncologic outcomes.
Author Response
In their retrospective, single-center study, the authors compared perioperative and oncologic outcomes of VATS single segmentectomy (SS) versus multiple segmentectomy (MS) in 334 patients with cT1N0 NSCLC ≤3 cm. The authors concluded that SS provides comparable oncologic control to MS, while offering better perioperative recovery. The topic is clinically relevant and timely, especially following recent randomized trials supporting parenchyma-sparing resections. The manuscript is generally well-structured and clearly written. However, several methodological and interpretative issues require clarification before the manuscript can be accepted.
Comment 1: The abstract is well written. However, the authors used several abbreviations without spelling them out the first time they are mentioned (e.g., OS, DFS). All abbreviations should be written in full upon first use in both the abstract and main text.
Response 1:Thank you for noting this oversight. We have now spelled out overall survival (OS), disease-free survival (DFS), and all other abbreviations at first mention in both the Abstract and the main text.
Comment 2: The authors provided the ethics committee approval number; for transparency, the exact date of approval should be mentioned.
Response 2: We agree and have now included the date of ethics approval: “approved on 5th June 2025” (line 115)
Comment 3: The authors analyzed patients up to December 2022, but no rationale is provided for ending data collection at this point, despite the lack of minimum follow-up criteria in the inclusion/exclusion criteria. If postoperative follow-up duration is important for interpreting outcomes, then a minimum follow-up period (e.g., ≥24 or ≥36 months) should be clearly stated as an inclusion requirement. Otherwise, the authors should explain why patients operated after December 2022 were not included. Since treatment strategies and tumor characteristics may change over time, this could introduce temporal selection bias, which should be acknowledged in the Limitations section.
Response 3: We thank reviewers for this comment. We now clarify that we ended data collection in December 2022 to ensure a minimum potential follow-up of 24 months to the 31th December 2024. We added this point in the methods (line 196). “Patient inclusion ended in December 2022 to ensure at least 24 months of potential follow-up to 31th December 2024”. (line 155)
Comment 4: Although the baseline characteristics are generally similar between groups, the choice between SS and MS was made intraoperatively based on achieving clear margins. This naturally introduces selection bias toward more favorable tumors in the SS group. The authors briefly mention this in the discussion but should expand on it. Specifically, they should clarify the institutional protocol for segment selection. They should also include a propensity score matching or inverse probability weighting analysis to reduce confounding. At a minimum, they should present a multivariable Cox regression model for DFS and OS.
Comment 4: We appreciate the reviewer’s attention to potential selection bias. We agree that such bias is an important consideration in retrospective comparative studies. However, in our cohort, the SS and MS groups were well balanced across baseline clinical and tumor characteristics, including age, sex, comorbidities, pulmonary function, tumor size, tumor location, histology, PET SUV, and C/T ratio (Tables 1 and 2). There were no statistically significant differences in these variables, and thus no identifiable confounders requiring adjustment. Because of this baseline comparability, performing propensity score matching would lead to a substantial reduction in sample size, without providing meaningful new information, and potentially introduce artificial imbalance by overcorrecting a balanced dataset. Similarly, multivariable Cox regression would not add explanatory value, as no variables met criteria for confounding. No changes.
Comment 5: The main issue of this study is the very short follow-up period. The median follow-up time is 30 months, which is insufficient to fully assess oncologic equivalence, especially regarding local recurrence. The authors should include landmark survival estimates at five years or more if available and clearly acknowledge that late recurrences might be missed.
Response 5: We thank the reviewer for highlighting this important point. We agree that early-stage NSCLC may recur beyond 3 years, and therefore a median follow-up of 30 months is insufficient to fully assess long-term oncologic equivalence, particularly with respect to local recurrence. Our survival curves currently represent the maximum available follow-up for this cohort at the time of data analysis. The proportion of patients with ≥5-year follow-up is limited, and generating 5-year survival estimates would introduce substantial statistical uncertainty. For this reason, we did not include extrapolated survival estimates or incomplete 5-year Kaplan–Meier outputs.
This point is acknowledged in the limitation section. We also revised the text in the conclusion by adding: “ While SS and MS demonstrated comparable short-term oncologic outcomes in this study, these findings should be interpreted with caution due to the relatively short follow-up period. Longer-term data are required to confirm durable oncologic equivalence”(line 389)
Comment 6: The authors did not clearly specify the primary or secondary outcomes of the study. Clearly defining the outcomes is essential for proper interpretation and scientific rigor. Outcomes of the study should be presented as a separate paragraph in the methodology.
Response 6: We thank the reviewer for this comment. We have now clearly defined the study outcomes in the Methods. The primary outcomes were overall survival (OS), disease-free survival (DFS), and local recurrence-free survival (LRFS). The secondary outcomes included perioperative complications, operative time, drainage duration, length of hospital stay, lymph node yield, and margin distance. This has been added in the Methods section: “line 147-153)
Comment 7: Tables – Every abbreviation used in a table should be explained in the table's legend.
Response 7: All abbreviations have now been defined in each table legend.
Comment 8: The authors stated that SS achieved larger median margins than MS, which is counterintuitive. They should specify the margin-to-tumor size ratio, as recommended in segmentectomy guidelines. Additionally, they need to discuss whether frozen section was used routinely, as it is currently stated as not routine, which conflicts with achieving larger margins.
Response 8: We thank the reviewer for this comment. We agree that reporting the margin-to-tumor size ratio is more informative than raw margin distance, in accordance with segmentectomy guideline recommendations. We have now calculated and added the margin-to-tumor ratio for both groups and included it in Table 2.
We also clarify that SS was predominantly performed for peripheral tumors located within a segment, where anatomical boundaries allow for clear visualization and shaping of a generous parenchymal margin along a single intersegmental plane. In contrast, MS was more frequently required for tumors located close to segmental borders, where achieving adequate margins necessitates removal of adjacent segments, but the intersegmental planes are more complex and irregular, probably resulting in narrower measured margins despite a larger volume of resection. Regarding frozen section, we confirm that frozen section was used selectively when margin adequacy was uncertain. In most SS cases, the tumor’s peripheral location within the anatomical limits of a single segment allowed the surgeon to obtain adequate margins without requiring intraoperative margin verification. We have now clarified this point in the discussion.
Discussion: “The fact that SS demonstrated a slightly larger median parenchymal margin (13 vs 11 mm; p=0.038) and margin-to-tumor ratio (0.9 vs 0.7; p=0.027) compared with MS may appear counterintuitive, but this finding may probably reflect anatomical selection rather than more extensive resection. SS was primarily performed for tumors located well within the boundaries of a single anatomical segment, where the intersegmental plane is regular, and the resection margin can be shaped generously. In contrast, MS was more often required when tumors were positioned near intersegmental borders, where multiple planes intersect, and the resection margin is constrained by neighboring bronchovascular structures. Therefore, even though MS removes a greater volume of lung tissue, the achievable margin may be smaller due to the irregularity of intersegmental boundaries and the need to preserve essential vascular and airway anatomy.” (line 320-330)
Comment 9: A primary reason for segmentectomy is tissue preservation. However, there is no postoperative pulmonary function data or quality-of-life metrics included. This should be recognized as a significant limitation and discussed more clearly.
Response 9: We thank the reviewer for this important comment. We agree that postoperative pulmonary function and quality-of-life outcomes are highly relevant in the context of parenchyma-sparing surgery. Unfortunately, standardized postoperative pulmonary function tests and patient-reported outcome measures were not routinely collected for all patients during the study period, and therefore could not be analyzed retrospectively.
We acknowledge that the absence of postoperative pulmonary functions and quality-of-life data limits our ability to directly quantify the functional advantage of single segmentectomy. We had already mentioned this limitations : “Sixth, functional and quality-of-life outcomes were not evaluated. These data would be essential to support the potential parenchymal-sparing benefits of single segmentectomy from a patient-centered perspective”. (Line 376)
Comment 10: The distribution of simple versus complex segmentectomies varies between groups. Since surgical difficulty is closely linked to operative time and complications, the authors should perform subgroup analyses based on complexity categories.
Response 10: We thank the reviewer for this comment. We agree that surgical complexity may influence perioperative outcomes. Although a subgroup analysis based on complexity could be performed, the study was not powered to detect meaningful differences within these smaller strata.Therefore, we believe a subgroup analysis would not add robust clinical insight and may generate misleading conclusions. However, we added a sentence in discussion about the high proportion of complex segmentectomie in single segment: .” In our cohort, a considerable proportion of SS involved segments traditionally considered technically demanding, such as S1, S3, S8, or S9, where intersegmental planes are frequently deep and irregular. Despite this higher anatomical complexity, perioperative outcomes (including air leak, complications, and length of stay) were comparable between SS and MS suggesting that complexity alone does not necessarily translate into increased morbidity”. (line 338)
Comment 11: A median follow-up of 30 months is too short to draw definitive conclusions about oncologic equivalence, especially regarding local recurrence. Early-stage NSCLC often shows recurrences after more than 3 years. Thus, stating that SS and MS offer comparable oncologic control is premature and should be rephrased to indicate that the findings are preliminary. The authors should clearly indicate that a longer follow-up period is necessary to verify the oncologic outcomes.
Response 11: We thank the reviewer for this important comment. We agree that a median follow-up of 30 months is relatively short for assessing long-term oncologic equivalence, especially since recurrences in early-stage NSCLC may occur beyond 3 years. Therefore, our conclusions have been revised to emphasize that the oncologic results represent short-term outcomes, and we have clearly stated that longer follow-up is required to confirm durability. Corresponding modifications have been made in the Abstract and Conclusion sections.
Round 2
Reviewer 2 Report
Comments and Suggestions for Authors
The authors have thoroughly revised the manuscript in accordance with the my comments, and I believe that the paper is now in a well-completed form. However, I would like to request only a simple minor revision.
Specifically, the information provided in Response 3—including the proportions of lobectomy and segmentectomy—should be summarized and presented in a supplemental table.
Author Response
Comment 1: The authors have thoroughly revised the manuscript in accordance with the my comments, and I believe that the paper is now in a well-completed form. However, I would like to request only a simple minor revision.
Specifically, the information provided in Response 3—including the proportions of lobectomy and segmentectomy—should be summarized and presented in a supplemental table.
Response 1: We thank the reviewer for this suggestion. To improve clarity, we have now included a supplementary figure (Supplementary Figure S1) illustrating the annual proportion of lobectomies and segmentectomies performed at our institution from 2017 to 2022. A brief explanation has been added to the Results section. The figure also highlights the progressive increase in segmentectomy over lobectomy during this period, reflecting the evolving surgical paradigm following growing evidence supporting parenchyma-sparing resections. We believe this visual summary provides the requested information in a concise and accessible manner.
We added in the results: “The annual distribution of segmentectomy and lobectomy procedures performed at our institution between 2017 and 2022 for cT1 N0 NSCLC is shown in Supplementary Figure S1. Over this period, segmentectomy progressively increased and ultimately surpassed lobectomy in early-stage NSCLC.” Line 245
Reviewer 3 Report
Comments and Suggestions for Authors
The authors have sufficiently addressed many of the requested revisions, and numerous corrections have clearly enhanced the clarity and structure of the manuscript. However, several key issues still remain inadequately resolved, and further refinement of the responses is necessary before the revision can be considered final.
- The response to Comment 4 is the most problematic. The decision to skip any adjusted analysis is not methodologically sound, as observed baseline balance does not rule out the possibility of confounding, especially in a context where the extent of resection is determined intraoperatively and influenced by anatomical and technical factors that are not fully captured in the dataset. The original review explicitly requested propensity score matching, inverse probability weighting, or at minimum a multivariable Cox regression, and this request has not been fulfilled. This should be addressed; otherwise, I will not provide positive feedback.
- The explanation of institutional decision criteria for choosing between single and multiple segmentectomy is also incomplete and needs clearer description.
- The response to Comment 10 remains only partially addressed. Although the manuscript might not be powered to detect statistically significant differences in subgroup analyses, the initial request did not call for formal hypothesis testing. An exploratory descriptive comparison between simple and complex segmentectomies would be informative and would improve transparency without overstating the findings.
- Several minor points could also be improved. The discussion of follow-up limitations would benefit from explicitly noting that five-year survival estimates are unreliable due to very small numbers at risk, clarifying why such estimates were not included.
- Additionally, the limitation regarding the absence of postoperative functional and quality-of-life data could be expanded to indicate whether these metrics are currently being collected prospectively.
Author Response
The authors have sufficiently addressed many of the requested revisions, and numerous corrections have clearly enhanced the clarity and structure of the manuscript. However, several key issues still remain inadequately resolved, and further refinement of the responses is necessary before the revision can be considered final.
Comment 1: The response to Comment 4 is the most problematic. The decision to skip any adjusted analysis is not methodologically sound, as observed baseline balance does not rule out the possibility of confounding, especially in a context where the extent of resection is determined intraoperatively and influenced by anatomical and technical factors that are not fully captured in the dataset. The original review explicitly requested propensity score matching, inverse probability weighting, or at minimum a multivariable Cox regression, and this request has not been fulfilled. This should be addressed; otherwise, I will not provide positive feedback.
Response 1: We sincerely thank the reviewer for this important recommendation. In response, we have now performed a comprehensive propensity score matching (PSM) analysis to minimize baseline confounding and to provide an adjusted comparison between SS and MS.
Propensity scores were calculated using a multivariable logistic regression model that included nine baseline covariates: age, smoking status, Charlson Comorbidity Index, comorbidities, FEV1, tumor size, consolidation-to-tumor ratio, involved lobe, and tumor location. Patients undergoing SS and MS were then matched in a 1:1 ratio using nearest-neighbor matching with a caliper width of 0.3 standard deviations of the logit of the propensity score. Covariate balance was assessed using standardized mean differences (SMD), with an SMD < 0.1 interpreted as adequate balance. Survival outcomes (OS, DFS, and LRFS) in the matched cohort were evaluated using Kaplan–Meier curves and log-rank tests.
The PSM procedure resulted in a matched cohort of 148 patients (74 SS and 74 MS). After matching, all covariates demonstrated excellent balance (SMD < 0.1 for all variables). In the matched analysis, no significant differences were observed between SS and MS in overall survival (HR 0.49, 95% CI 0.09–2.74; p = 0.414), disease-free survival (HR 0.74, 95% CI 0.16–3.42; p = 0.700), or local recurrence-free survival (HR 1.10, 95% CI 0.47–2.60; p = 0.826). Postoperative morbidity and mortality were also similar between groups after matching.
Overall, the PSM analysis supports our primary findings. After adjustment for key confounders, SS and MS demonstrated comparable long-term oncologic outcomes. Full methodological details and the complete results of the PSM analysis are presented in Supplementary Tables S1–S5.
We added in the methods: “To reduce selection bias, we performed a sensitivity analysis using propensity score matching (PSM). Propensity scores were estimated using a multivariable logistic regression model including nine baseline covariates: age, smoking status, Charlson Comorbidity Index, comorbidities, FEV1, tumor size, consolidation-to-tumor ratio, involved lobe, and tumor location. Patients undergoing SS and MS were matched 1:1 using nearest-neighbor matching with a caliper of 0.3 SD of the logit of the propensity score. Covariate balance was assessed using standardized mean differences (SMD), with <0.1 indicating adequate balance.
Survival outcomes (OS, DFS, LRFS) in the matched cohort were evaluated using Kaplan–Meier curves and log-rank tests. PSM results are provided in Supplementary Tables S1–S5.” (Line 219)
We added in the results section: “The propensity score matching analysis produced a matched cohort of 148 patients (74 SS and 74 MS). Baseline characteristics were well balanced across all nine covariates (SMD < 0.1). In the matched cohort, OS (p = 0.414), DFS (p = 0.700), and LRFS (p = 0.826) did not differ significantly between SS and MS. A full summary of PSM-adjusted survival outcomes is presented in Supplementary Table S4.” (line 290)
We added in the discussion: “The PSM sensitivity analysis further confirmed that the type of segmentectomy was not associated with differences in OS, DFS, or LRFS after balancing nine clinically relevant confounders. (line 367)”
Comment 2: The explanation of institutional decision criteria for choosing between single and multiple segmentectomy is also incomplete and needs clearer description.
Response 2: Thank you for this valuable comment. We have clarified the criteria used at our institution to determine whether a tumor is managed with single segmentectomy (SS) or multiple segmentectomy (MS).
In our instituion, the extent of segmental resection is decided preoperatively based on thin-slice CT evaluation and confirmed intraoperatively. The key determinant is the anatomical relationship between the tumor and segmental/intersegmental planes, with the goal of achieving a parenchymal margin ≥ 2 cm or ≥ the tumor diameter. SS is preferred when the tumor is clearly located within a single anatomical segment, allowing a safe and oncologically adequate resection along one intersegmental plane. MS is selected when the tumor is adjacent or crossing a segmental boundary, when adequate margin cannot be achieved within one segment.
We added in the methods: “The decision to perform single (SS) or multiple segmentectomy (MS) was based on preoperative thin-slice CT assessment and intraoperative evaluation of segmental anatomy. SS was performed when the tumor was clearly confined within a single segment and a parenchymal margin ≥ 2 cm or ≥ tumor diameter could be safely achieved along one intersegmental plane. MS was chosen when the tumor was located near or across a segmental border, when achieving an adequate margin within one segment was not feasible.“ line 179
Comment 3: The response to Comment 10 remains only partially addressed. Although the manuscript might not be powered to detect statistically significant differences in subgroup analyses, the initial request did not call for formal hypothesis testing. An exploratory descriptive comparison between simple and complex segmentectomies would be informative and would improve transparency without overstating the findings.
Response 3: We thank the reviewer for this clarification. We fully agree that an exploratory descriptive comparison between simple and complex segmentectomies would improve transparency without implying formal statistical significance. In response, we have now added a propensity score–matched (PSM) descriptive comparison of perioperative characteristics (operative time, drainage duration, length of stay, and postoperative complications) stratified by simple versus complex procedures. These data are presented in Supplementary Table S3.
Comment 4: Several minor points could also be improved. The discussion of follow-up limitations would benefit from explicitly noting that five-year survival estimates are unreliable due to very small numbers at risk, clarifying why such estimates were not included.
Response 4: We thank the reviewer for this suggestion. We agree that clarifying the reason for not presenting five-year survival estimates strengthens the discussion of follow-up limitations. We have now added a sentence explicitly noting that the number of patients at risk beyond 48 months was very small, making five-year survival estimates statistically. We added in the limitations: “Due to the limited number of patients remaining at risk beyond 48 months, five-year survival estimates would be statistically unreliable and therefore were not reported.” (Line 392)
Comment 5: Additionally, the limitation regarding the absence of postoperative functional and quality-of-life data could be expanded to indicate whether these metrics are currently being collected prospectively.
Response 5: We thank the reviewer for this comment. We agree that the limitation regarding the absence of postoperative pulmonary function and quality-of-life data should be clarified. We have now expanded the text to specify that these metrics were not systematically collected during the study period and are not currently recorded in a standardized prospective manner at our institution. As a result, functional comparisons between SS and MS could not be performed. We adapted in the limitations: “Sixth, postoperative pulmonary function tests and quality-of-life assessments were not systematically collected during the study period and are not currently obtained in a standardized prospective manner at our institution. Consequently, we were unable to evaluate the functional impact of SS versus MS, which represents an inherent limitation.” (Line 403)
Round 3
Reviewer 3 Report
Comments and Suggestions for Authors
Thank you for your thorough revision. The newly added PSM analysis, clarification of resection criteria, and expanded discussion of study limitations satisfactorily address all major points raised in the previous review. The supplementary descriptive comparison is appropriate as presented. I have no further comments, and the manuscript is suitable for publication.